# What Do Chinese Entrepreneurs Think about Entrepreneurship: A Case Study of Popular Essays on Zhisland

**Zhenping Zhang [1] , Haiyan Yan [2],* and Jiayin Qi [3]**

[1]  Department of Political Sciences and International Relations, University of Palermo, 90100 Palermo, Italy; zhenping.zhang@unipa.it
[2]  School of Management, Shanghai University of International Business and Economics, 1900 Wenxiang Road, Songjiang District, Shanghai 201600, China
[3]  Institute of Artificial Intelligence and Change Management, Shanghai University of International Business and Economics, 1900 Wenxiang Road, Songjiang District, Shanghai 201600, China; ai@suibe.edu.cn
*  Correspondence: yhy@suibe.edu.cn

**Abstract:** Entrepreneurship is becoming increasingly essential in this current era of the knowledge economy. It contributes to the innovation of products and services as well as improved processes. In the long-run, it can also improve the sustainability of the economy by depicting better efficiency and social goals. To stimulate entrepreneurship, it is essential to investigate the thinking behind entrepreneurship or what entrepreneurs think about entrepreneurship. Such investigations should encompass the mental images of entrepreneurs. In this regard, content analysis, based on the popular Zhisland essays, may be applied to elicit opinions from Chinese entrepreneurs about activities, critical factors and intended outcomes within the ambit of entrepreneurship. In this study, 634 concepts are first coded and categorized into 20 second-level themes and six first-level themes. The six first-level themes are competing strategy, human resource management, management and leadership, marketing and sale, research and development, and risk management. Furthermore, among the 20 second-level themes, leadership, self-improvement, and the risk of business cycle attract the highest attention, each of which accounts for around 10% of the coded concepts. Finally, a causal loop diagram is depicted to synthesize the coding results. This study also underscores three essential activities of entrepreneurship, which entail building and maintaining competing advantage, improving user experience, and risk management. Entrepreneurs need to balance investments in those activities according to the change in environment and customers' needs.

**Keywords:** entrepreneurship; entrepreneur; content analysis; causal loop diagram; system dynamics; mental image

---

## 1. Introduction

With the increase in the pervasiveness of the knowledge economy, innovation is increasingly being recognized as the new engine of the economy, which calls for a high level of entrepreneurship. Entrepreneurship is the process of designing, launching, and running a new business, which is often initially a small business, while an entrepreneur is an entity that seeks for and acts upon opportunities to transform inventions or technologies into products and services. In China, entrepreneurship has played a pivotal role in the development and transformation of the economy from quantity to quality, and from manufactory to innovation, which has been well recognized [1].

Given that entrepreneurship plays a significant role in the development and transformation of the economy, various measures have been taken to stimulate the development of entrepreneurial activities

in China. For instance, entrepreneur education and various training programs have been promoted to train potential entrepreneurs from inside and outside universities [2]. In addition, many competitions in the development of business plans are held occasionally. During such competitions, entrepreneur experts and managers participate in the evaluation of the quality of program proposals. Later awards and funding are then allocated to accelerate the growth of start-ups. Chen, Yao, and Kotha (2009) investigated the extent to which venture capitalists' (VCs') perceptions of "entrepreneurial passion" influence the VCs' investment decisions [3]. They found that it was preparedness, not passion, that positively impacted decisions to fund ventures. In this regard, when the start-ups are built, it is essential to provide mentor services, finance, and preferential tax so as to enhance the developing process of entrepreneurship [4].

Although such strategies are strongly promoted with preferential policies and activities, the participation rate is still low because of a lack of identifiable role models, poor media presentation of individuals or small firms, lack of encouragement from important influencers on career choices [5], and uncertainty and risk. Thus, entrepreneurship is deemed by the public as risky, difficult, and mysterious. Therefore, it is necessary to investigate what entrepreneurs think about entrepreneurship, based on which targeted policies may be implemented to cultivate entrepreneurship.

This paper therefore aims to investigate Chinese entrepreneurs' mental image of entrepreneurship, which entails the assumptions and causal models that guide their decision-making processes. To achieve this goal, three research questions need to be asked. First, what are the critical activities of entrepreneurship in the Chinese entrepreneurs' view? Second, how do they prioritize the critical activities of entrepreneurship? Finally, how do those activities connect?

This paper begins with a summary of the literature regarding entrepreneurship development and critical success factors. Next, the research methodology and design are introduced and applied to elicit the mental image of entrepreneurs from popular essays on social media. The coding results and a drawn causal loop diagram are summarized and discussed. Finally, it concludes with the findings and suggestions for future research.

## 2. Literature Review

There are currently various discussions on entrepreneurs and entrepreneurship in the economic and management literature. According to Schumpeter and Opie (1961), an entrepreneur is a person who is willing and able to convert new ideas or inventions into a successful innovation. Entrepreneurship employs what they called "the gale of creative destruction" to replace, in whole or in part, inferior innovations across markets and industries, simultaneously creating new products or new business models to earn a profit or to strengthen competing advantages [6]. Deakins and Freel (2009) define entrepreneurs as leaders who are willing to take risks and exercise initiative, taking advantage of market opportunities by planning, organizing, and deploying resources, often by innovating to create new or improving existing products or services [7].

While entrepreneurship is often associated with new, small, for-profit start-ups, entrepreneurial behavior can be seen in small-, medium- and large-sized firms. It may also transcend to new and established firms and in for-profit and not-for-profit organizations, which includes private businesses, voluntary sector groups, charitable organizations, and government. While for-profit entrepreneurs typically measure performance using business metrics like profit, revenues, the return of investment, and increases in stock prices, social entrepreneurs are either non-profits or blend for-profit goals with generating a positive "return to society", and therefore rely on the application of different metrics, such as public welfare, happiness, and quality of life. Social entrepreneurship typically attempts to further broad social, cultural, and environmental goals often associated with the voluntary sector [8]. Similarly, Yun (2015) concluded three kinds of innovation which entail open innovation, closed innovation, and social innovation [9]. Such a perspective has created a new way to understand the dynamics of entrepreneurial activities and competing strategy. Furthermore, recent research also investigates the well-being of entrepreneurs, which has been defined as the experience of satisfaction,

positive affect, infrequent negative affect, and psychological functioning concerning developing, starting, growing, and running an entrepreneurial venture [10].

Other studies have also investigated the uncertainty of entrepreneurship and the effect of entrepreneurs' personality. Entrepreneurship is often associated with uncertainty and risk, particularly when it involves the creation of a new good or service, for a new market that did not previously exist, rather than when a venture brings an incremental effort to improve an existing product or service. Entrepreneurs are often overconfident and adventurous, and exhibit an illusion of control, when they are opening/expanding business or new products/services [11]. However, much as overconfidence can encourage the employees and enable the fuzzy missions, it may create a situation where the entrepreneurs overestimate the development of the business and overlook the potential risk. An entrepreneur typically has a mindset that seeks out potential opportunities during uncertain times and makes full use of it before it is discovered by competitors [12]. For instance, which making use of the survey data on 98 CEOs at public art performance centers, Kim and Jung (2015) confirmed the positive effect of the characteristics of the chief executive officer (CEO), such as entrepreneurship, social responsibility (perception), and social capital, on management outcomes, with quality management activities as the intermedia variable [13].

Entrepreneurs are supposed to acquire a high level of leadership, which is heavily influenced by the existing cultural context and path-dependent. For instance, the participative leadership style that is encouraged and practiced in western countries is considered disrespectful and inappropriate in many other parts of the world due to the differences in the power structure and distance [14]. Many Asian and Middle Eastern countries do not have "open door" policies for subordinates, which means that supervisees would never informally approach their managers/bosses to participate in the high-stakes decision-making process. In such countries, an authoritarian approach to management and leadership is more legitimate and customary. Besides, the development of both the market and traditional cultures also influence the entrepreneurs' perception of the relationships and rules. For example, as the market environment is not fully developed, the market is closely regulated by the government and highly influenced by informal relations. Guanxi (personal connection) has been identified as a necessary condition to do business successfully in China [15]. He et al. (2019) reviewed the research of entrepreneurship in China [16]. They concluded that one of the most important dimensions of informal institutions is associated with China's traditional culture, which plays a decisive role in shaping value orientations and entrepreneurs' cognition. The psychological characteristics of entrepreneurs, such as their risk-taking propensity, are especially heavily induced by the general cultural context, and China's "Confucian values and ethics" in particular. Liu et al. (2018) investigated the relationship between religion and entrepreneurship in China [17]. They argue that entrepreneurs' association with Buddhism helps to build social and political capital, to grow corporate social responsibility, and to adopt a long-term business vision. Their empirical analysis of 1032 young Chinese firms supports the positive relationship between having a Buddhist entrepreneur and young firms' sales and profit.

The current research shows that the concept of entrepreneurship has evolved from launching a new business to any entrepreneurial activities that involve a new way to operate a business or NGO, or an existing organization. For the critical factors, previous research has examined the personality of leaders and the motivations for entrepreneurship. However, few of them systemically examine the critical factors. They also seldom examine these factors from the perspective of entrepreneurs. This paper, therefore, aims to fill this gap by investigating Chinese entrepreneurs' mental image of entrepreneurship and systemically synthesizing their knowledge. The novelty of this paper is three-fold. First, the knowledge about entrepreneurship is extracted from popular essays from social media, which are well recognized by entrepreneurs and written in their language. Next, a causal loop diagram is drawn to synthesize the knowledge systematically, connecting the critical factors, activities, and outcomes of entrepreneurship with causal relations. Finally, the cultural and research context is examined carefully in the analyzing process to strengthen the findings and to appropriately generalize the findings to the other contexts.

## 3. Materials and Methods

### 3.1. Research Methodology

The System Dynamics (SD) methodology was found by the pioneering work of Forrester (1961) [18], which aims to investigate complex systems through a cause-and-effect perspective. This preliminary analysis provides a basis to build simulation models oriented towards supporting decision-makers' learning processes. In fact, through the investigation of simulation results derived from the implementation of desired policies, decision-makers can acquire a deeper knowledge of the relationship between the cause-and-effect system structure and the related main variables' behaviors over time. It was first applied in supply chain management. Nowadays, it is widely used in social science, engineering, healthcare and education, with applications in public, private, and non-profit organizations [19].

To adapt to different situations and modeling objectives, modelers can look at the system in various ways. At the institutional level, performance is assessed primarily on the effects produced by decision-makers on their institutions without considering external stakeholders. At the interinstitutional level, performance is assessed based on the effects of decision-makers on the wider system, either a local area or the industry to which they belong [20]. Such effects may include tax contributions, increasing employment levels, protected environment, and sharing information, knowledge and resources with business partners, among others. Besides, there are three types of models, conceptual model, stock-and-flow model, and insight model. The conceptual model depicts the feedback loops explaining system behavior, with no quantitative data or simulation. The stock-and-flow model is quantitative, which implies a level of detail, and accuracy, and an extension of explored system boundaries, i.e., usually broader than in qualitative analysis. An insight or policy-based model is built based on its usefulness to users, rather than how closely it mimics the reality of historic time series. Therefore, relatively simple models might be valid, and just as effective as a highly detailed model [20–22].

The strengths of System Dynamics include: thinking dynamically, thinking in stocks and flows, thinking in feedback loops, and thinking endogenously [23]. Thinking endogenously refers to the effort to see the cause of the system as being internal forces rather than outside intervention, to extend the boundary of thinking and connect the cause and effect in a consistent map. Besides, not only does System Dynamics build models based on numerical data but it also emphasizes the importance of mental models, which entails the assumptions and causal mental models that guide the managers' decision-making process.

In this paper, the concepts and causal relations are interpreted from the essays, thus the objective is to get an overview of the system rather than to do numerical simulation. The conceptual model will be used to depict the mental image of entrepreneurs. The conceptual model, that is to say, the causal loop diagram, consists of variables and links, while variables are central concepts and links depict the cause-and-effect relations. Figure 1 describes the causal relations among the population, birth rate, and death rate. The variable "Population" has a positive effect on both the birth rate and the death rate. Then, the birth rate has positive feedback and the death rate has a negative feedback on "Population". Finally, two feedback loops are depicted: one is a positive feedback loop formed by "Population → birth rate → Population"; the other one is a negative feedback loop formed by "Population → death rate → Population". An arrow with a positive sign means positive causal relations, while an arrow with a negative sign means a negative relation. Similarly, a positive loop is a loop that exaggerates what happens, while a negative loop counteracts the change, which pushes the system back to the original state before the change.

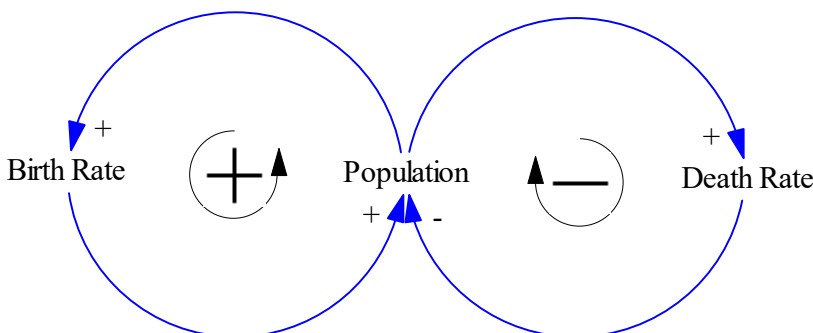

**Figure 1.** An example of a causal loop diagram.

*3.2. Data Collection and Analysis Process*

To elicit the mental image, capturing the entrepreneurs' opinions on entrepreneurship was essential. It involved the use of Zhisland, which is a popular social platform for Chinese entrepreneurs, where users post opinions and essays, and have discussions with others via comments and likes. Most of the users are influential entrepreneurs who are certificated based on their affiliation and their position in their organization. There are four levels of certification, with the top one being categorized for top managers in big companies or founders of middle-scale companies. There is also a popular ranking in the Zhisland app, in which users can recommend essays to the platform while others respond through comments or likes. The ranking is calculated based on the comments and likes received, and it is updated every 10 min from 9 p.m. on the first day to 9 p.m. on the second day. Each day, 100,000 RMB is awarded to the authors and the commenters of the three most popular essays, and the recommenders of the top 20 essays. Zhisland then uses these incentives and ranking mechanisms to select the most popular essays every working day, which are recognized by the Chinese entrepreneurs.

The ranking was initially released on 25th July 2019. Since then, three essays have been chosen every day from the ranking. This study collected the most prominent essays from 1st August to 31st August; in total, 93 essays were collected. This study applied three standards during the process of essay selection. First, the topic had to be connected to entrepreneurial activities. Second, the selected essays had to be rich in content and discussion. Finally, the selected essays had to cover entrepreneurs in different industries, including manufacturing, retailing, and e-commerce among others. Following these standards, nine essays were chosen for further analysis. The titles and related information of the nine essays can be seen in Appendix A.

A three-step coding scheme is applied, i.e., open codes, axial codes, and selective codes [24]. First, a few concepts were derived from the data based on open codes, that is to say, it was extracted in "a grounded way". Then, based on the content, the study classified the extracted concepts into different thematic categories according to axial codes. A two-level thematic analysis was conducted because there were many concepts. Lastly, a causal loop diagram was developed to synthesize the entrepreneurs' mental image of entrepreneurship. While the first two steps were carried out using Nvivo, the causal loop diagram was developed with Vensim.

## 4. Results

First, the study introduces the results of coding. In total, 634 concepts were extracted based on open codes, which includes the critical point of the essays. Then, the concepts are reviewed to be categorized according to general themes. As depicted in Table 1, the open codes are classified into six thematic categories, which entail competing strategy, human resource management, management and leadership, marketing and sale, research and development, and risk management. This demonstrates that the entrepreneurs' opinions cover almost every aspect of organization management, including macro-political and economic environment, industry development, inner organizational management, and outside market competition. Furthermore, risk management, human resource management,

and management and leadership are emphasized, each of which accounts for around 20% of the coded concepts.

**Table 1.** The first level of thematic coding.

| Theme | Concepts Count | % of Coded Concepts |
|---|---|---|
| Competing strategy (CS) | 94 | 14.8 |
| Human resource management (HRM) | 136 | 21.5 |
| Management and leadership (M&L) | 122 | 19.2 |
| Marketing and sale (M&S) | 56 | 8.8 |
| Research and development (R&D) | 72 | 11.4 |
| Risk management (RM) | 154 | 24.3 |
| Total | 634 | 100% |

Then, the concepts are reviewed for a second time, drawing out secondary themes, of which are 20. As shown in Table 2, leadership, self-improvement, and business cycle attracted the highest attention, each of which accounts for around 10% of the coded concepts. In the competing strategy, self-improvement is highlighted, which consists of organizational change and technological improvement to compete with benchmarks. Environment and trend monitoring is placed before cooperation, as the business environment of China is changing rapidly and closely regulated by the government. In human resource management, the four sub-themes share almost equal percentage, while recruitment is slightly emphasized. The young generation is quite different from the previous generation, because they are more well-off and independent than previous generations. Therefore, Chinese entrepreneurs make a big effort to understand and learn the characteristics of young people. In management and leadership, besides the traditional way of acting as an example to lead and manage, the importance of formal rules and work processes is also recognized. In marketing and selling, user needs and buying experience are highlighted, as consumers gain more power than previously. Besides marketing and selling, additional service is provided to satisfy customers' needs. For example, some enterprises provide free delivery services and cookies. In research and development, both technology and customer understanding are important. While customer understanding guides the direction, the development part requires more investment in research and development employees and advanced equipment. In risk management, four kinds of risk are emphasized, which entail business cycle, natural disaster, social revolution, and technology advancement, among which the business cycle seems the most important.

Based on the thematic analysis, a causal loop diagram is drawn to synthesize the entrepreneurs' view of entrepreneurship. First, for each second-level theme, the core concepts and outcomes are drawn, then the links are drawn based on the causal relationship among them. For example, human resource management is connected to management and leadership, while research and development is a part of user experience improvement.

As Figure 2 shows, there are three critical factors to achieve the success of entrepreneurship. They are competing advantage, user experience, and effect of risk and crises, which, in turn, are driven by six reinforcing loops. R1 is driven by cooperation with external organizations, such as industry association, government, industry leaders, and allies. R2 is about the environment and trend monitoring, which contributes to the improvement in management efficiency. R3 is a combination of HRM and management, as enterprise motivates and leads young people to act as a team and evolve as the environment changes. The first three loops help to build and maintain a competing advantage. R4 focuses on risk management, which prepares for uncertainty and risk. R5 and R6 aim to improve user experience by marketing and research and development. Customer understanding is critical for both service improvement and product development. These reinforcing loops depict the driving

engine of entrepreneurship. Finally, these six loops are interconnected. For example, environment and trend monitoring also contribute to the preparation for risk and crises, and management efficiency helps to quicken the process of product development.

**Table 2.** The second level of thematic coding.

| 1st Level Theme. | 2nd Level Theme | Concepts Count | % of Coded Concepts |
|---|---|---|---|
| CS | cooperation | 12 | 1.9 |
| CS | environment and trend monitoring | 20 | 3.2 |
| CS | self-improvement | 62 | 9.8 |
| HRM | evaluation and treatment | 34 | 5.4 |
| HRM | recruitment | 49 | 7.7 |
| HRM | training | 31 | 4.9 |
| HRM | understanding of new-generation people | 22 | 3.5 |
| M&L | leadership | 71 | 11.2 |
| M&L | management | 51 | 8.0 |
| M&S | additional service | 2 | 0.3 |
| M&S | marketing | 35 | 5.5 |
| M&S | selling | 19 | 3.0 |
| R&D | customer understanding | 22 | 3.5 |
| R&D | development | 25 | 3.9 |
| R&D | new technology | 25 | 3.9 |
| RM | business cycle | 60 | 9.5 |
| RM | definition and concept | 27 | 4.3 |
| RM | natural disaster | 11 | 1.7 |
| RM | social revolution | 39 | 6.2 |
| RM | technology advancement | 17 | 2.7 |
| | Total | 634 | 100% |

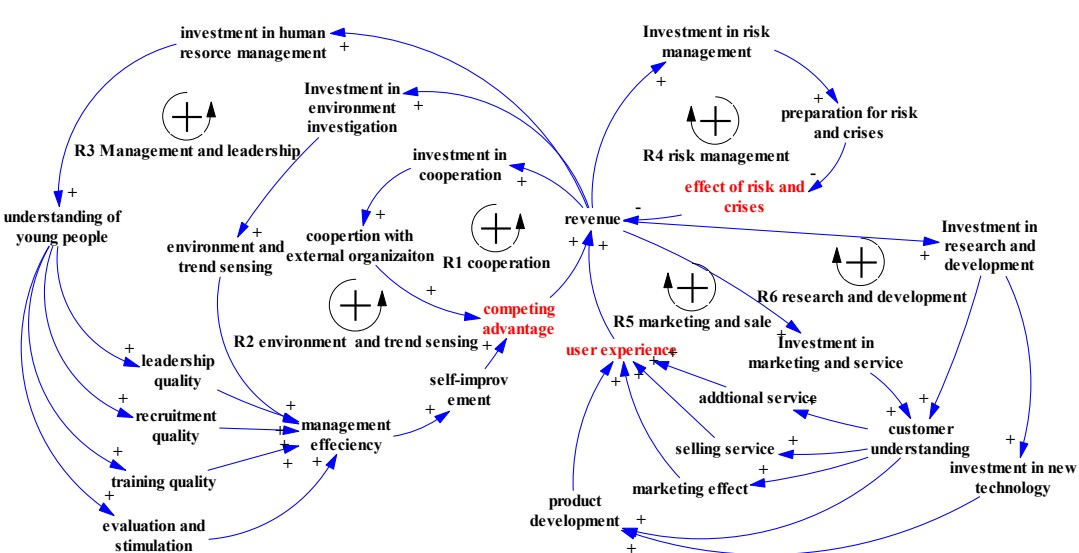

**Figure 2.** A Causal Loop Diagram of entrepreneurs' view of entrepreneurship (Part I reinforcing loops).

For the sake of brevity and readability, the balancing loops are drawn separately in Figure 3. The investments in different loops share the same pool of revenue. Therefore, entrepreneurs need to balance the investment in cooperation, environment and trend monitoring, management and leadership, risk management, marketing and sale, and research and development. Running entrepreneurship is like driving a car, and the six kinds of investments are wheels of the car, which need to be developed in balance.

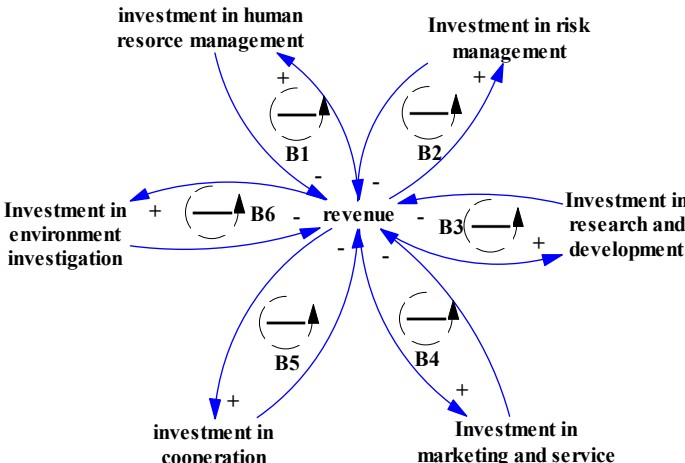

**Figure 3.** A Causal Loop Diagram of entrepreneurs' view of entrepreneurship (Part II balancing loops).

## 5. Discussion

The coding result shows that entrepreneurs are facing complex issues, ranging from team-building and leadership to the outside environment and trend-monitoring. This is consistent with the current literature, which characterizes entrepreneurship as uncertain and challenging, requiring high-level skills of communication, leadership, marketing, and trend-monitoring. With the upgrading of the Chinese economy, user experience and cooperation with allies are emphasized as necessary to survive in the changing environment. As the young generation is quite different from the previous ones, being independent, well-off, and familiar with new technology, Chinese entrepreneurs highlight the importance of understanding and changing the ways of working with young employees. Besides, as user experience is vital for entrepreneurs' success, customer understanding becomes a critical part of research and development and marketing. Finally, risk management is regarded as an indispensable part of entrepreneurship to handle the business cycle, natural disaster, social revolution, and technology advancement.

With the help of the causal loop diagram, we can frame entrepreneurship in a consistent and connected map. There are three activities: management and leadership to build and maintain competing advantage, risk management to survive the crises, and marketing and research and development to improve user experience. These three activities are interconnected. On the one hand, they facilitate each other by sharing a knowledge base and work processes. On the other hand, all of them drain out the pool of revenue. Enterprise needs to balance investments in different parts according to the change in the environment and customers' needs, which reinforces the view of Simon (1996), who argued that the enterprise can be seen as an artifice, which is designed by its goal. It continually adjusts the inner environment to adapt to the outside environment [25]. Similarly, to monitor the dynamics of the market, demand articulation is emphasized as a critical factor [26]. Entrepreneurs need to balance different kinds of investments and actions to satisfy customer requirements.

One of the critical research questions in our paper is how entrepreneurs prioritize factors and activities. Normally, in the stock and flow model, the weight of different factors can be measured by their influence on the outcomes through simulation. Although the weight cannot be directly calculated from the qualitative coding, the frequency of coded concepts in the essays can be a good indicator of their weights. In this sense, among the six first-level themes, risk management, human resource management, and management and leadership have higher weights than the other three. However, in the 20 s-level themes, self-improvement, leadership, and business cycle are the three most important factors.

Strangely, the prominent guanxi of Chinese management research is missing in the essays. Does the environment change so much that the guanxi doesn't matter anymore? To some extent, the answer is yes. This is because as the government reform towards the market economy continues, the market environment becomes fair and the competition process becomes transparent. However,

there are other reasons why guanxi does not appear in the essays. First, all the essays are written by successful entrepreneurs who are running big companies. They are competent and do not need to rely on preferential policies. Chen and Chen (2004) differentiated the development of guanxi into three sequential stages: initiating, building, and using [15]. For those successful entrepreneurs, they have built enough guanxi to support their business and guanxi is not their focus of entrepreneurship. Second, the guanxi is a part of informal rules, which are tacit and not suitable for sharing the related experiences of guanxi in public. In Chinese culture, achieving success by guanxi is not fair and can attract criticism and blame from the public. Finally, it is also connected to the time frame. Guanxi is a complex issue that takes a lot of time and resources to build and maintain. Maybe it is not the short-term critical factor, while it still is for long-running businesses, which require continuous long-term investment.

This study's contribution to theory is two-fold. First, it elicits the mental image of Chinese entrepreneurs from popular essays in a systemic way, which adds to the knowledge of entrepreneurship in China. As the market environment is changing rapidly, both the employees and customers are changing too, and entrepreneurs need to modify their ways of working with young people and update the direction of product development in a timely manner. Second, although the result confirms the evolving nature of entrepreneurship and the importance of entrepreneurs' leadership, it differs from the literature in the description of guanxi, which highlights the importance of the research context, as this study investigates successful entrepreneurs' speaking in public. Therefore, future research should specify the stages of entrepreneurship, and whether it is done in public or private, to strengthen the validity of their research findings.

This study has several practical implications. First, as the new generation is quite different from the previous ones, entrepreneurs need to learn more about their employees and also change their management style. Such changes should entail building a free and flexible working environment to attract creative employees. As the coding results suggested, when monetary stimulation does not work, some managers try to leverage employees' passion and interest and improve the charm of leaders. Second, as the work division among different sectors become more embedded in the industry, cooperation has been a considerable competing strategy to deal with risk and uncertainty together. As Zeng and Chen (2003) suggested, entrepreneurs need to manage the inherent tension between cooperation and competition in alliances [27]. The six critical activities facilitate each other but also share the same pool of revenue. Therefore, entrepreneurs need to balance investments in cooperation, environment, and trend monitoring, management and leadership, risk management, marketing and sale, and research and development, according to the change in the environment and customers' needs. Finally, to cultivate entrepreneurship, the government or universities can provide help with those six activities. For example, the customer report can be provided to help potential entrepreneurs find profitable markets. When the environment changes rapidly, reminders and suggestions may be sent to protect the entrepreneurs from potential risks.

A shortcoming of qualitative research is that it cannot quantify the relationship between factors and outcomes. Indeed, the relation can be measured in the stock-and-flow simulation model. Structural dominance analysis can be applied to examine the effect of different factors over time [28]. In this paper, only popular essays on Zhisland are analyzed and the processes of coding and diagram drawing are subject to personal selection and bias. Future research can also use other supplementary methods, such as interviews, observation, and action research, to collect the data in different ways and quantify the elicited causal loop diagram to measure the relative effects of different factors and loops. We also treat the essays as a whole without discrimination; future research can compare the mental images of different entrepreneurs. It should be noted that this study is about successful entrepreneurs' speaking in public; future research can also investigate entrepreneurs' thinking in other contexts.

## 6. Conclusions

This study depicts the influential role and importance of entrepreneurship in an economy. Entrepreneurship contributes to the development of the economy as entrepreneurs develop innovative

products and services to satisfy customer needs. The current research treats the entrepreneurs' personality and critical factors as isolated, without examining the context as well as the environment, and their internal connections. In this paper, the popular essays on the entrepreneur-oriented social media were collected to elicit the entrepreneurs' mental image of entrepreneurship. This included their understanding of the critical factors, activities, and outcomes of entrepreneurship. It was found that there are three critical activities of entrepreneurship. The first one is to maintain a competing advantage over competitors, which requires a perception of the environment and future trends, recruiting and stimulating the young generation, and cooperating with external organizations. The second activity is to improve user experience, either by marketing and sales or by investing in research and development, both of which need a deep understanding of customers. The final activity is risk management, especially the risk caused by the business cycle and advancements in technology. All three activities share the same pool of revenue, therefore entrepreneurs need to balance investments in different parts according to the change in the environment and customer needs.

**Author Contributions:** Formal analysis, Z.Z. and J.Q.; Investigation, Z.Z.; Methodology, J.Q.; Supervision, H.Y.; Writing—original draft, Z.Z.; Writing—review & editing, H.Y. All authors have read and agreed to the published version of the manuscript.

**Funding:** This research received no external funding.

**Acknowledgments:** The authors thank Chuanchao Sun from Shanghai University of Finance and Economics for his support of data collection. Besides, the authors also thank Donald Fitz Omong from University of Palermo for his help to improve the language.

**Conflicts of Interest:** The authors declare no conflict of interest.

## Appendix A

This is the related information of the nine selected essays.

**Table A1.** The title and relative industry of the selected essays.

| No. | Title | Industry |
|---|---|---|
| 1 | Li Lin: Anyone who wants to do retailing should learn this to succeed | Retailing |
| 2 | Wei Ze: Rewards and punishments should be clear, and heavy punishment should be used in troubled times! | E-commerce |
| 3 | Feng Lun: Reflect on how to survive by learning the four ways of death of an enterprise | Real estate |
| 4 | Liu Donghua: There is only one way to Rome: seek the cause in yourself | Media |
| 5 | Chen Kaixuan: from zero to 20 billion in 25 years, 8 pieces of my life beliefs | Consumer goods |
| 6 | Dong Minzhu: Gree is fighting all the way out | Electric appliance |
| 7 | Wang Shi: whether an enterprise can succeed or not depends on the selection of personnel | Real estate |
| 8 | Deng Kangming, former chief human resources officer of Alibaba: the core team is left over, not selected | E-commerce |
| 9 | Ren Zhengfei: the best defense is attack, we will dominate the world | Communication devices |

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
