# Peer review of "What Do Chinese Entrepreneurs Think about Entrepreneurship: A Case Study of Popular Essays on Zhisland"

_2199-8531, doi:10.3390/joitmc6030086_

Round 1
Reviewer 1 Report
Dear Authors,
Thank You for the opportunity of reading this article. My general opinion about this article is correct. I decided to suggest the accepting this article after minor revision.
General statements about the article:
-> The problem that is presented in this article is actual and desirable. This topic suites to Journal of Open Innovation: Technology, Market, and Complexity journal.
-> The abstract of the article is adequate for its content. Its length is correct.
-> Organization of the article is correct. It contains 5 sections that are ordered in an adequate manner.
-> The presented in figure 2 and 3 loops diagrams are a strong point of the article.
-> The discussion part is complex and sufficient.
-> Quality of figures and tables is sufficient.
However, I indicate some elements that require revisions:
#1 Novelty of the article
Generally in the introduction, the novelty of this work should be emphasis. Lines 101-107 should be extended in point of highlighting the novelty of this article.
#2 Literature review
Generally, the literature review in this article is sufficient. However in my opinion there is a lack of indication of similar research. I think it would be desirable to improve by adding 3-4 positions that present similar research from the last 2-3 years.
#3 decimal part
Please uniform the decimal part of “% of coded concepts” in table 2. E.g. in place of “8” should be “8.0”.
#4 abbreviations
I think that list of abbreviations would increase the legibility of the article.
Some specific remarks
-> line 111 -> decimal should be in the previous line
-> line 324 -> I’s suggested to not use abbreviations in the conclusion section. So please use the whole name of R&D.
Author Response
Thank you for your careful reading and kind suggestions.

Reviewer 2 Report
In order to express more precisely it is recommended to thoroughly revise some statements or expressions e.g:
verse 5-7: "(...) entrepreneurship (...) contributes to new products and services, improved processes and creative innovation and even the sustainability of the economy."
This sentence should be rewritten for 3 reasons:
1) new products & services + improved processes = innovations. That is why statement "new products & services and improved processes and innovations" is illogical;
2) each innovation is supposed to be "creative". Instead "creative innovation" just "innovation" is better;
3) not every kind of entrepreneurship contributes directly to sustainability of the economy. This statement requires extra assumptions.
verse 27: Instead:"(...) translate inventions (...) into products, I recommend "transform";
verse 321-322: Instead "sensing the environment and future trend", I recommend "monitoring";
For improving the clarity of fig. 2 I recommend to highlight the 3 main success factors of entrepreneurship.
My comment on the statements contained in verses 267-278: It is worth to investigate (in further research) whether guanxi is no longer the critical success factor at all? Maybe it is not only as the short-term critical factor while is still for the long-run businesses.
Author Response

(The authors gave the same response as above.)

Reviewer 3 Report
Authors have classified several concepts and their each other relations and their relations to entrepreneurship but they didn't provide a "wheight" for each connection they found. This could say how much one concept is important compared to another...
It may be interesting to include this kind of measure, or, at least, an interesting theoretical and literature subject to mention and to refer to, as a way to deepen and better explain the connections.
Author Response

(The authors gave the same response as above.)
